# Diagnosis of Slipped Capital Femoral Epiphysis: How to Stay out of Trouble?

**DOI:** 10.3390/children10050778

**Published:** 2023-04-25

**Authors:** Vito Pavone, Gianluca Testa, Paola Torrisi, Kathryn Louise McCracken, Alessia Caldaci, Andrea Vescio, Marco Sapienza

**Affiliations:** 1Department of General Surgery and Medical Surgical Specialties, Section of Orthopaedics and Traumatoogy, A.O.U.P. Policlinico Rodolico—San Marco, University of Catania, 95123 Catania, Italy; gianpavel@hotmail.com (G.T.); paolatorrisi50@gmail.com (P.T.); alessia.c.92@hotmail.it (A.C.); andreavescio88@gmail.com (A.V.); marcosapienza09@yahoo.it (M.S.); 2School of Medicine, University College Cork, College Road, T12 K8AF Cork, Ireland; 120105096@umail.ucc.ie

**Keywords:** slipped capital femoral epiphysis, SCFE, delay, diagnosis, risk factors

## Abstract

Slipped capital femoral epiphysis (SCFE) is the most common hip disorder affecting children and adolescents aged between 9 and 16 years, affecting approximately 10 per 100,000 children per year. The diagnosis of SCFE is often delayed, leading to an increased risk of complications. This study aims to provide the latest evidence concerning the causes of diagnostic delay and risk factors for SCFE and to educate general practitioners and paediatricians to help reduce delays in diagnosis and provide earlier therapeutic intervention. A literature search was conducted in the ScienceDirect and PubMed databases according to the PRISMA statement. Suitable studies for this systematic review included 22 articles discussing the aetiology of SCFE, risk factors, and causes of late diagnosis. Causes of delayed diagnosis include underestimation by patients, initial diagnostic approach by a non-orthopaedic professional, inadequate imaging, failure to recognize morphological changes, and variation in symptomatic presentation. The underlying risk factors for SCFE are likely part of a multifactorial process which involves anatomical variations and the metabolism of leptin, growth hormone, insulin, and other metabolic parameters. This review highlights the importance of early recognition and diagnosis of SCFE and proposes an algorithm for physicians to approach children who may have this condition.

## 1. Introduction

Slipped capital femoral epiphysis (SCFE) is the most common hip disorder affecting children and adolescents aged between 9 and 16 years. SCFE occurs due to posterior and medial translational movement of the epiphysis relative to the metaphysis as a result of a weakened proximal femoral physis [1]. The annual incidence of SCFE is approximately 10 cases per 100,000 children [2]. A highly appropriate aetiology theory is probably one with multiple factors. Because SCFE only happens during puberty, when many hormonal changes take place and physeal longitudinal growth reaches its peak, it may be caused by a combination of mechanical forces that stress the body and produce slippage. The most frequent presentation is pain around the hip, groin or thigh. Many patients report referred pain from the knee. Clinical examinations may reveal leg length disparity and poor gait. Internal rotation is reduced when comparing the hip range of motion to the contralateral hip and external rotation may be increased. A diagnostic finding is the Drehmann’s sign, which refers to abduction and external rotation of the hip joint in response to passive flexion. Regular radiography is used to confirm and classify SCFE diagnoses. The hip is typically assessed using two radiographs, the anteroposterior (AP) and frog-leg views. The final one is performed while supine, with the knees flexed (30–40°), the hips abducted (45°), and the foot soles together and externally rotated. If, however, the patient exhibits symptoms, a body habit, and a physical examination compatible with Slipped Capital Femoral Epiphysis, but results of Rx-view are negative, Magnetic Resonance Imaging (MRI) may identify physeal widening and irregularity, which are the initial indicators of SCFE [3].

SCFE is often insidious and the diagnosis delayed. Despite being frequently associated with obesity, atypical SCFE still occurs in non-overweight individuals. The primary treatment option for SCFE is in situ pinning. Other treatments for severely unstable slips include osteotomies, open reduction, capsular decompression and hip arthroplasty [4]. An early diagnosis is crucial to avoid medium- and long-term complications requiring further treatment. According to a study conducted by the Mayo Clinic, avascular necrosis (AVN) is the main indication for management with hip arthroplasty in individuals with SCFE [5]. It is caused by blood vessel kinking or hematoma formation, which compromises the fragile blood supply to the femoral head and is linked with significant displacement or multiple screw fixation. Symptomatic therapy is used until the pain becomes excessive, at which point total hip arthroplasty is performed [6].

This procedure is extremely challenging for both patients and healthcare providers and is atypical in relatively young subjects, usually less than 50 years old [7], as they frequently require further revision procedures. In Denmark, 0.5% of hip arthroplasties are a consequence of SCFE, but this value is likely underestimated, due to the high rate of missed SCFE diagnosis [8].

The aim of this study is to provide the latest evidence concerning the causes of diagnostic delay and risk factors for the disease. Furthermore, this study aims to educate general practitioners and paediatricians on the high prevalence of SCFE and to help reduce delays in diagnosis and provide earlier therapeutic treatments which will in turn decrease the risk of complications.

## 2. Materials and Methods

### 2.1. Selection Criteria

This study was conducted according to the PRISMA statement [9] (Appendix A, Preferred Reporting Items for Systematic Reviews and Meta-Analysis), through a systematic review of the Science Direct and PubMed databases, concluding on 30 June 2021. It included a 27-item checklist and a flow chart divided into 4 steps. The checklist contains the essential items for transparent reporting of systematic reviews. Suitable studies included those discussing the aetiology of SCFE, risk factors, and causes of late diagnosis. The initial selection of titles and abstracts was made using the following inclusion criteria: (1) randomized controlled trials (RCT) studies, prospective studies (PS), retrospective studies (RS), case control studies (CC), and cohort studies (C); (2) studies written in the English language; (3) studies reporting clinical or preclinical results; (4) content closely related to the purpose of the review study. The study excluded articles about contralateral Slipped Capital Femoral Epiphysis. Other exclusion criteria included out-of-topic articles, articles with a low level of evidence based on the ROBINS-I tool for non-randomized studies and those without an accessible abstract and full text. 

### 2.2. Literature Search Strategy and Data Extraction

The search string included the terms “Slipped capital femoral epiphysis OR SCFE OR epiphysiolysis” with “Clinical OR Radiological OR X-ray OR neglect or misdiagnosis OR imaging OR delay” AND “Diagnosis” AND “Prognosis OR risk factor OR indicator”, either as MeSH terms or keywords. Two investigators (PT and AV) independently selected and reviewed each article, its texts, tables, and figures. The following data were extracted for analysis: the characteristics of the study, including the first author, year of publication, number of patients, number of SCFE cases, and the clinical characteristics of patients. Cases of discrepancies between the two reviewers were mostly resolved by consensus following discussion. The senior surgeon reviewed any remaining conflicts. The intended outcome of this systematic review was to find the latest evidence concerning the causes of diagnostic delay and risk factors for the disease.

### 2.3. Assessment of Bias Risk

In this systematic review, the bias risk evaluation was carried out using the ROBINS-I tool for non-randomized studies: it consists of a three-step assessment. The first step concerns the initial planning of the systematic review. The second step is evaluating the common biases that can be found in these studies. The third step concerns the overall bias risk. Two authors (AV and PT) carried out the evaluation independently. Any discrepancies were discussed with the senior researcher (GT) for the final decision. All evaluators agreed on the final decision of each assessment step (Table 1 and Table 2).

## 3. Results

After searching the literature, 881 articles were found. After the exclusion of duplicates, 426 articles were selected. At the end of the first screening, we selected 146 articles eligible for reading the whole text following the selection criteria described above. Finally, after checking the reference lists, 22 eligible articles were selected for the final analysis. A PRISMA flowchart of the selection and screening method is provided (Figure 1).

### 3.1. Anatomical Factors

In 2020 Morris et al. [10], in an X-ray cohort study, reported that SCFE patients had lesser contralateral epiphyseal cupping compared to controls: superiorly (*p* < 0.001) and anteriorly (*p* < 0.001). The cupping is measured as the ratio of the femoral head diameter to the difference between a line parallel to the femoral neck axis and the height of the epiphysis at its center. A decrease in the epiphyseal extension ratio was also reported compared to controls (*p* < 0.01).

### 3.2. Body Weight

Manoff [11], Song [12], Madhuri [13] and Perry [14] conducted case-control studies which all found a significant difference between the Body Mass Index (BMI) of the epiphysiolysis group compared to the control group (*p* > 0.0001; *p* = 0.0078; *p* = 0.006) or compared to the British reference population. Obana et al. [15] evaluated the BMI of 275 children with SCFE and demonstrated how “normal-weight” SCFE patients are mainly found in older (*p* = 0.015), female (*p* = 0.034) patients, and present with a more severe (*p* = 0.007) and unstable (*p* = 0.001) slip than overweight and obese SCFE patients.

### 3.3. Metabolic Disorders

Halverson et al. [16] evaluated the role of leptin in relation to body weight through four study groups: obese and non-obese patients with SCFE showed higher leptin levels than the two control groups (*p* = 0.006; *p* = 0.039). Furthermore, after incorporating all variables through a multivariate logistic regression, the study reported a significantly higher chance of SCFE diagnosis based solely on leptin elevation, regardless of obesity status, sex, and race (OD = 4.9; 95%CI:1.31–18.48; *p* < 0.02). Montanez-Alvarez et al. [17] analysed serum levels of glucose, insulin, glycated haemoglobin, lipid profile, and complete blood count. The study reported significantly higher scores of insulin resistance (*p* = 0.005), hyperinsulinemia (*p* = 0.005), triglycerides (*p* = 0.037), and VLDL (*p* = 0.009) in patients with SCFE compared to the control group. Taussig et al. [18] evaluated BMI and blood pressure levels. The study reported a significantly lower BMI in SCFE, as well as a significantly higher percentage of patients with hypertension in the SCFE group (*p* < 0.01).

### 3.4. Bone Dysmetabolism

Madhuri et al. [13] showed significantly decreased vitamin D levels (mean of 11.78 ng/mL) in SCFE patients but no significant correlation between these levels and slip angles (r = 0.0). Papavasiliou et al. [19] reported a high incidence of serum abnormalities in PTH, Ca, and P levels in the SCFE group compared to patients in the control group.

### 3.5. Hormonal Disorders

Papavasiliou et al. [20] conducted a cohort study that revealed SCFE patients were more likely to display a temporary hormonal disturbance of T3, T4, TSH, GH, testosterone, DHEA, estradiol, ACTH, FSH, LH and cortisol rather than a true endocrinopathy. De Andrade et al. [21] measured Southwick’s angle before and during treatment with growth hormone (GH) in 44 patients with stature deficit: no significant differences were found between these patients and the control group, before and after one year of treatment (*p* = 0.3). Furthermore, no correlation was assessed between Southwick’s angle and puberty (*p* = 0.7), gender (*p* = 0.8), bone age (*p* = 0.9), BMI (*p* = 0.2), or other hormonal deficiencies (*p* = 0.4).

### 3.6. Socio-Economic Deprivation

Perry et al. [14] demonstrated a significant association between socio-economic deprivation and SCFE incidence (95%CI:1.05–1.21; *p* < 0.001) in their cohort study.

### 3.7. Late Diagnosis

#### 3.7.1. Radiological Assessment

Billing et al. [30] studied how the slipping angle (SA), according to the geometric method, was a more accurate diagnostic method than the conventional method (AP and FL view) (*p* < 0.05) in their case-control study. Three studies highlighted the diagnostic accuracy of the frog projection [22,23,31], up to 100% sensitivity. “Trethowan sign” is evaluable in anteroposterior (AP) and frog-leg lateral (FL) view: the sign is positive when Klein’s line, a line extending along the superior femoral neck, does not intersect the physeal region [23]. In the FL view, the S-sign is a curved line drawn along the proximal femoral physis on the inferior border of the proximal femoral head-neck junction. Any breaks in the S-continuity, asymmetry or sharps turns were noted as abnormal tests that indicated SCFE [22]. Lastly, on the frog lateral hip view, Southwick slip angle is the most used measure of SCFE severity: it is defined as the angle formed by the shaft and a line perpendicular to the epiphysis [31] (Figure 2).

#### 3.7.2. Misleading Symptoms

Uvodich et al. [24] reported that solitary hip pain was the most common pain pattern (*n* = 35), followed by combined hip and thigh pain (*n* = 10) and hip and knee pain (*n* = 9). Less commonly reported pain patterns were groin and knee (*n* = 1), groin, thigh and knee (*n* = 1) and posterolateral pain (*n* = 1). The study reported a greater slip angle in hips with knee pain than those without (*p* = 0.018) and in subjects with combined pain than in those with isolated pain (*p* = 0.003). Lastly, the authors noted that rare pain presentations had a more significant duration of symptoms (*p* = 0.039) and visits until diagnosis (*p* = 0.036). From 2004 to 2021, several studies reported that patients with predominantly knee pain took a significantly longer time to reach diagnosis than patients with hip pain: Kocher et al. [25] (15.0 weeks vs. 6.0 weeks; *p* < 0.001), Schur et al. [26] (30.6 weeks vs. 15.7 weeks; r = 0.16; *p* < 0.001), Perry et al. [14] (*p* < 0.001; 161 vs. 20 days), Hosseinzadeh et al. [27] (*p* = 0.0097; HR = 0.6; 95%CI0.4–0.9) and Ortegren et al. [28] (41 vs. 22 weeks; *p* < 0.003).

#### 3.7.3. Patient Presentation Delay

Hosseinzadeh et al. [27] showed that the time between the onset of symptoms to the patient’s first medical examination was longer, but not significantly so, for visits with an orthopaedic specialist (median = 91 days) compared to a non-specialist (median = 27 days) (HR = 1.465; *p* = 0.051). Furthermore, Ortegren et al. [28] reported a significantly longer patient delay than the medical delay: 10 weeks (range 1–57) vs. 4 weeks (range 0–57) (*p* = 0.002).

#### 3.7.4. First Contact

Schur et al. [26] observed that patients assessed at an orthopaedic clinic (0 weeks) had a significantly reduced time from the first assessment to diagnosis compared to patients assessed by a family doctor (4 weeks; *p* = 0.003) or emergency room (6 weeks; *p* = 0.008). Hosseinzadeh et al. [27] reported similar evidence (*p* < 0.05) as well as a longer period of time from diagnosis to surgical treatment (*p* < 0.0001). These studies differ from Uvodich et al. [24], who reported no significant association between first medical contact and diagnostic delay time.

#### 3.7.5. Demographic Information

Ortegren et al. [28] found a significantly longer total delay in males (*p* = 0.038) compared to female patients (*p* = 0.021) but no significantly longer total delay when comparing the age of onset of symptoms (*p* = 0.47). Kocher et al. [25] reported no significant association between diagnostic delay and gender (*p* = 0.999) or between the age of onset and diagnostic delay (*p* = 0.078). However, there was a significant association between major delay and lower family income (*p* = 0.007), as well as between delay with Medicaid insurance versus private insurance (12 vs. 7.5 weeks). Hosseinzadeh et al. [27] reported significantly shorter time lags for Medicaid (93 days) or private insurance (72 days) compared to no insurance (162 days). However, these results conflict with some other studies in which no significant association was found with age at the first symptoms (*p* = 0.40), gender (*p* = 0.13), or type of insurance (*p* = 0.2) [24].

#### 3.7.6. Types of SCFE

Ortegren et al. [28] noted a significantly lower delay in diagnosis in patients with mild slip (<30%) than in patients with moderate or severe slips (>30%) (13 weeks vs. 29 weeks; *p* = 0.002); they also analysed lower slip angles for patients diagnosed within 20 weeks of onset. Schur et al. [26] found the opposite: a longer delay in patients with moderate to mild slips (r = 0.24; *p* < 0.001) and association of a greater Southwick angle with a longer delay (Spearman r = 0.31; *p* < 0.001). Kocher et al. [25] confirmed a significant relationship between a longer delay in diagnosis and greater slip severity (*p* < 0.013) and reported a significant association between a longer delay in diagnosis and stable slips (*p* < 0.10). Schur et al. [26] reported a longer time from the first symptom to diagnosis in stable SCFE (r = 0.24; *p* < 0.001) and for patients with an initial bilateral presentation (*p* < 0.001).

## 4. Discussion

In some cases, the diagnosis of SCFE can be difficult. It is often delayed due to the atypical patterns or blurred symptomatology. Despite being frequently associated with obesity, atypical SCFE still occurs in non-overweight individuals. A highly appropriate aetiology theory is probably one with multiple factors. Because SCFE only happens during puberty, when many hormonal changes take place and physeal longitudinal growth reaches its peak, it may be caused by a combination of mechanical forces that stress the body and produce slippage. The most frequent presentation is pain around the hip, groin or thigh. Many patients report referred pain from the knee. Clinical examinations may reveal leg length disparity and poor gait. Internal rotation is reduced when comparing the hip range of motion to the contralateral hip and external rotation may be increased. The knowledge of the causes of delay and risk factors for SCFE are important for a correct patient classification, early diagnosis, and for limiting complications and medico-legal issues; Danish patients have been able to file a claim through the independent Danish Patient Insurance Association (DPIA) when they experience an unanticipated side effect or harm as a result of their medical treatment [32].

Although there is no clear and universally accepted definition of delayed diagnosis for SCFE, three studies shown that approximately 63% of patients are diagnosed after 4 weeks from the onset of symptoms [14,23,29]. One of the main causes of late diagnosis is the underestimation of the clinical picture by patients. Male patients come later to medical observation, as do those with mild or moderate slip severity [28].

Another cause of delayed diagnosis is an initial diagnostic approach by a member of a non-orthopaedic medical profession, such as physiotherapists, chiropractors, primary care physicians, and family doctors [26,27]. Only 19% of diagnoses are made within 1 week by non-orthopaedic health care professionals compared to 97% of diagnoses made within 1 week when seen by orthopaedic surgeons [27]. The delay could be justified by a wide variation in symptomatology; in fact, up to 11 different pain patterns have been identified. The most common pain pattern was solitary hip pain, followed by hip and thigh pain combined, and hip and knee pain. The most common atypical pain was pain in the thigh alone, followed by isolated knee pain, and then isolated groin pain [24]. It is usually mild and vague, and it can be intermittent or continuous, with exercise aggravating it. Pain might last for weeks or months after it first appears [33].

Furthermore, other causes of delayed diagnosis may include: an inappropriate instrumental procedure, such as the absence of a frog projection in order to avoid exposure to ionising radiation [23] or a failure to recognise morphological changes such as epiphyseal cupping [10]. Doctors should prescribe specific procedures to diagnose Slipped Capital Femoral Epiphysis.

Regarding predisposing factors, a combination of skeletal dysmorphisms and an increased load could play an important role in SCFE aetiology. Historically, the typical SCFE patient was a hypogonadal, obese adolescent boy [33]. Obesity increases the sheer force across the physeal plate, which causes repeated trauma to the physis and eventually causes the epiphysis to separate from the metaphysis. Although studies have linked obesity to an increased risk of SCFE, no research studies have examined whether there is a BMI limit beyond which a child’s risk for SCFE increases [11]. Litchman et al. [34] hypothesised that anatomical abnormalities such as femoral and acetabular retroversion, and excessive acetabular coverage in combination with excessive loading, may generate sufficient shear forces to cause the slip. The hypothesis has subsequently been supported by several studies [11,12,14,35] but this literature review analysis shows that in a total of 613 children [11,12,15] with SCFE, 299 (48.77%) were obese and 314 (51.22%) non-obese.

Therefore, obesity must be only one factor of a more complex multifactorial process [16] involving the metabolism of leptin, growth hormone, insulin, hypertension, and with Homeostatic Model Assessment (HOMA), triglycerides and VLDL as additional elements. Specifically, leptin would directly affect its own receptors in the physis, causing a dose-dependent diameter expansion of the proliferative zone [36].

Using a similar mechanism, GH treatment in patients with GHD could expose them to an increased risk of SCFE. The hormone may act directly on the growth cartilage and consequently increase local IGF-1 production and clonal proliferation. Indirectly, GH triggers chondrocyte proliferation in the physis by stimulating IGF-1 production in the liver [36]. Similarly, insulin, directly and indirectly, stimulates chondrocyte receptors, which in turn would cause the differentiation and proliferation of physeal chondrocytes. Hyperinsulinemia and increased Homeostatic Model Assessment scores [17] could explain the mechanical insufficiency of the physis to support high or repetitive loads during accelerated growth, leading to SCFE.

As demonstrated by Taussig et al. [18], increased blood pressure could act on the chondrocyte cell zone, generating a similar avascular necrosis picture. Some studies show that changes in bone metabolism could alter the action of parathormone (PTH) on growth cartilage chondrocytes resulting in aberrant mineralisation and prolongation of the time required for epiphysis fusion [19]. Calcitriol, an active form of vitamin D, acts on epiphyseal chondrocytes as well as displaying synergistic effects with PTH on chondrocyte proliferation [13].

Obana observed that patients with SCFE who are normal weight are more likely to be older, female, and with a severe and unstable SCFE. The older age may be related to the more rapid skeletal maturation frequently observed in obese patients. Skinnier patients may have initially been passed over for SCFE because of their low BMI, which led to a later diagnosis age and a progression to worsening slip. The severe and unstable SCFE in normal weight patients could be related to structural variation that could play a role in nonobese as well as obese patients. Further investigation is needed [15].

The primary strength of this systematic review is the extensive search and identification of all possibly relevant aetiologies. Therefore, the overall risk of bias for the review is low to moderate and likely did not influence the analysis. However, a major limitation of the review is that the literature available on the SCFE aetiology demonstrates great heterogeneity. Although many studies focused on the biomechanical, hormonal, and metabolic background of the disease, there is a lack of consensus on one or a few major actors responsible for the aetiopathogenesis. While obesity and anatomical factors seem to be associated with SCFE aetiology, more studies are needed to understand the complexity and multifactorial nature of SCFE pathogenesis. Another limitation of the study is the heterogenicity of the definition of SCFE diagnosis delay. Moreover, it was not possible to cover all parameters evaluated in the literature. Lastly, in order to include as many articles as possible, medium to low quality papers may have been included.

## 5. Conclusions

SCFE diagnosis is frequently delayed due to the typical versus atypical manifestation and misleading symptoms. In order to properly approach children who may have SCFE, a diagnostic algorithm should be followed. The proposed algorithm is shown below (Figure 3). The first step is to stratify patients according to hip and/or knee symptoms. If pain and typical manifestations are present, the second step is to perform imaging techniques (AP and FL view): Trethowan sign, Southwick angle, epiphyseal cupping, and the S-Sign have to be evaluated. If the pain and atypical manifestations are present, measurement of hormonal and metabolic parameters should be performed. The imaging and serum parameters will indicate appropriate further investigations or treatments.

The systematic review highlights the importance of the knowledge of the condition; this knowledge is necessary to enable health professionals to recognise SCFE and properly diagnose it. For this reason, educating general practitioners and paediatricians about the high prevalence of the disease and its risk factors must be a priority. Collaboration between physicians can improve early diagnosis, and a closer doctor-patient relationship can reduce delay in the diagnostic and therapeutic approach, thus reducing the risk of complications.

## Figures and Tables

**Figure 1 children-10-00778-f001:**
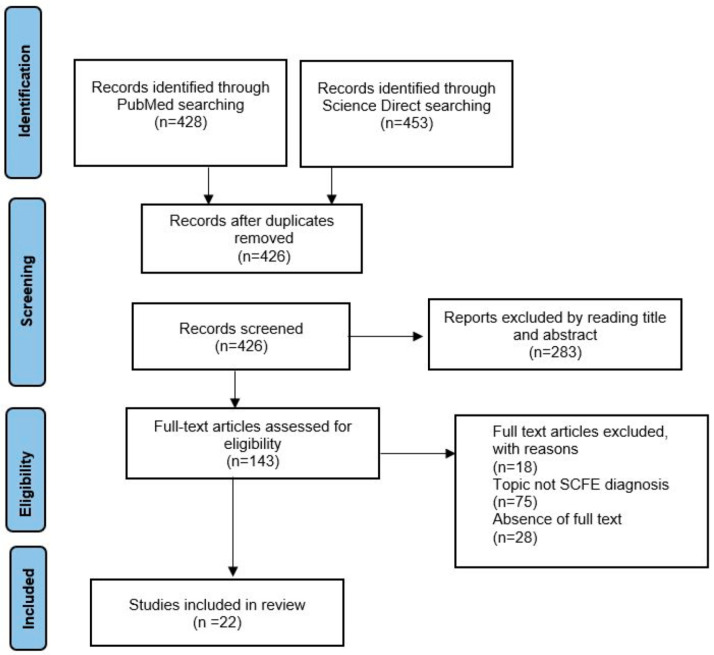
PRISMA (Preferred Reporting Items for Systematic Reviews and Meta-Analysis) flowchart of the systematic literature review.

**Figure 2 children-10-00778-f002:**
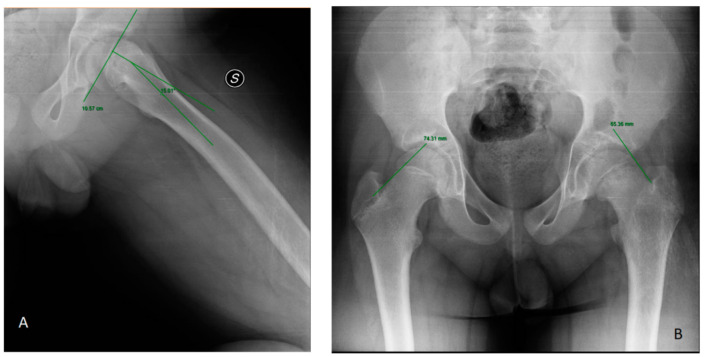
(**A**) Klein’s line. (**B**) Southwick slip angle in left SCFE.

**Figure 3 children-10-00778-f003:**
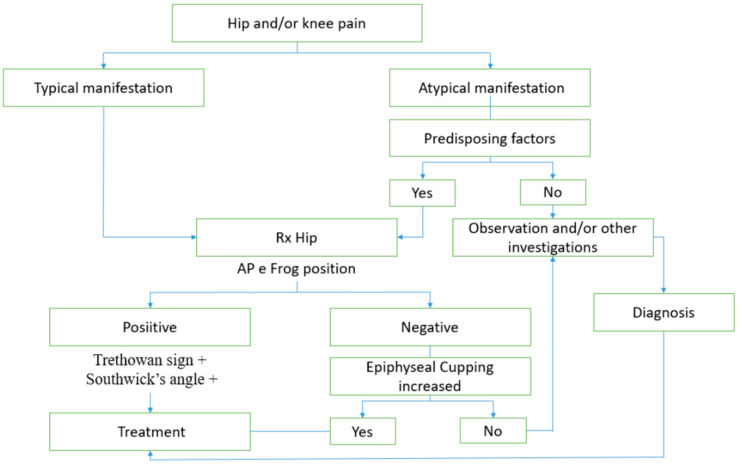
Diagnostic algorithm.

**Table 1 children-10-00778-t001:** Risk factors (HOMA: Homeostasis Model Assessment; VLDL: Very Low-Density Lipoprotein; LDL: Low-Density Lipoprotein; HDL: High-Density Lipoprotein; HbA1c: Glycosylated hemoglobin; BMI: Body Mass Index; GHD: Growth Hormone Deficiency).

Ref.	Subjects	Association
Morris et al. [10] (2020)	279 SCFE radiographs 279 Control radiographs	Anatomical factors
Manoff et al. [11] (2005)	106 patients with SCFE; 46 Controls	BMI, Gender
Song et al. [12] (2009)	231 patients with SCFE	BMI
Madhuri et al. [13] (2013)	15 patients with SCFE; 15 Controls	Vitamin D, BMI
Perry et al. [14] (2017)	596 patients with SCFE	BMI, Socio-economic deprivation
Obana et al. [15] (2020)	Normal weight (34 pz) Overweight (48 pz) Obese (193 pz)	BMI
Halverson et al. [16] (2017)	SCFE 40; Controls 30	Leptin
Montañez-Alvarez et al. [17] (2020)	SCFE 14 (3 overweight; 11 obese) Controls 23 (23 obese)	Hyperinsulinemia, HOMA, Triglycerides (VLDL, LDL, HDL) Total cholesterol Glycemia, Neutrophils, HbA1c
Taussig et al. [18] (2016)	127 patients with SCFE (obese 72%; overweight 14%) 127 Controls (obese 97%; overweight 3%)	Hypertension
Papavasiliou et al. [19] (2007)	A: 14 patients with SCFE B: 5 patients treated for SCFE	Parathormone, Calcium and Phosphorus
Papavasiliou et al. [20] (2007)	16 hips	Hormones
De Andrade et al. [21] (2009)	44 GHD patients	Southwick angle

**Table 2 children-10-00778-t002:** Late diagnosis factors.

Ref.	Subjects	Association
Rebich et al. [22] (2018)	35 patients with SCFE	Radiological evaluation
Samelis et al. [23] (2020)	36 stable slips	Radiological evaluation
Uvodich et al. [24] (2019)	107 patients with SCFE	Misleading symptoms, first contact
Kocher et al. [25] (2004)	196 patients with SCFE	Demographics characteristics Misleading symptoms
Schur et al. [26] (2016)	481 patients with SCFE	Misleading symptoms Demographic information First contact
Hosseinzadeh et al. [27] (2015)	149 patients with SCFE	Demographic characteristics, misleading symptoms, Type of medical provider, patient’s delay
Ortegren et al. [28] (2021)	54 patients with stable SCFE	Demographic information, misleading symptoms, first contact, patient’s delay
Rahme et al. [29] (2006)	102 patients with SCFE	First contact

## Data Availability

Not applicable.

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
