# Peer review of "Diagnosis of Slipped Capital Femoral Epiphysis: How to Stay out of Trouble?"

_children, 2023, doi:10.3390/children10050778_

Round 1

Reviewer 1 Report

It is an interesting topic. Here are some comments:

The abbreviation for slipped capital femoral epiphysis (SCFE) is stated twice in the introduction.

In the section “Literature search strategy”, the authors stated that “This study was conducted and concluded in June 2021, according to the PRISMA guidelines.”. Please state the day of the month. Please also note that PRISMA statement is a checklist for reporting of systematic reviews, not a methodological guideline. I suggest that the PRISMA statement is mentioned in the beginning of the method section (e.g., This review is reported in accordance with the PRISMA statement.).

Was a PRISMA checklist provided in the supplementary material?

In my opinion, the inclusion criteria should be mentioned before the sources of literature. This is because we first want to know what you were looking for and then where you were looking for it.

In the section “Literature search strategy”, please state the number of investigators that selected the studies and if they worked independently.

“references cited in systematic review reports on the same or similar topic” is mentioned as an inclusion criterion, but this is a source of literature.

”studies of any level of evidence” is mentioned as an inclusion criterion; however, review articles, and case reports are mentioned as exclusion criteria. Please state the types of papers that you were looking for in the list of inclusion criteria.

There is no need to explain what papers you were not looking for in the list of exclusion criteria because the papers would not be considered for inclusion in the first place. This just adds unnecessary text. Similarly, English language is mentioned as an inclusion criterion and non-English language is reported as an exclusion criterion.

Please do not sate the number of papers identified by the search in the method section, this is a result.

In the section “Selection criteria”, the authors also stated that “It has also excluded all the duplicates, out-of-topic articles, and those with poor scientific methodology or without an accessible abstract.”. Duplicates is not considered an exclusion criterion. Furthermore, it confuses me that studies of poor scientific methodology were excluded – this needs to be explained to readers. Also, I do not know why articles without an abstract were excluded – please explain.

In the discussion section, the authors stated that “The primary strength of this systematic review is the extensive search and identification of all possibly relevant etiologies. Therefore, the overall risk of bias for the review is low-moderate and likely did not influence analysis.”. I respectfully disagree that the literature search was comprehensive as it only included two databases and papers reported in non-English language was excluded. Experts in the field could have been contacted and asked to provide further published and unpublished studies and the reference lists of all the included studies should have been read. Also, there is no mention of the number of investigators that selected the studies.
Also, I disagree that a comprehensive literature search will always lead to a risk of bias of a review.

Author Response

Dear author, thank you for your important suggestions. Next you will find the review point by point.

Point 1:
In the section “Literature search strategy”, the authors stated that “This study was conducted and concluded in June 2021, according to the PRISMA guidelines.”. Please state the day of the month. Please also note that PRISMA statement is a checklist for reporting of systematic reviews, not a methodological guideline. I suggest that the PRISMA statement is mentioned in the beginning of the method section (e.g., This review is reported in accordance with the PRISMA statement.).

Was a PRISMA checklist provided in the supplementary material?

Response 1: We added and modified this point. Yes it was, we uploaded it in the supplementary material

Point 2: In my opinion, the inclusion criteria should be mentioned before the sources of literature. This is because we first want to know what you were looking for and then where you were looking for it.

In the section “Literature search strategy”, please state the number of investigators that selected the studies and if they worked independently.

“references cited in systematic review reports on the same or similar topic” is mentioned as an inclusion criterion, but this is a source of literature. ”studies of any level of evidence” is mentioned as an inclusion criterion; however, review articles, and case reports are mentioned as exclusion criteria. Please state the types of papers that you were looking for in the list of inclusion criteria.

There is no need to explain what papers you were not looking for in the list of exclusion criteria because the papers would not be considered for inclusion in the first place. This just adds unnecessary text. Similarly, English language is mentioned as an inclusion criterion and non-English language is reported as an exclusion criterion.

Please do not sate the number of papers identified by the search in the method section, this is a result.

In the section “Selection criteria”, the authors also stated that “It has also excluded all the duplicates, out-of-topic articles, and those with poor scientific methodology or without an accessible abstract.”. Duplicates is not considered an exclusion criterion. Furthermore, it confuses me that studies of poor scientific methodology were excluded – this needs to be explained to readers. Also, I do not know why articles without an abstract were excluded – please explain.

Response 2: We modified the Selection criteria and the order of the paragraphs. I meant, we excluded articles without an accessible abstract and full text associated.

Point 3: In the discussion section, the authors stated that “The primary strength of this systematic review is the extensive search and identification of all possibly relevant etiologies. Therefore, the overall risk of bias for the review is low-moderate and likely did not influence analysis.”. I respectfully disagree that the literature search was comprehensive as it only included two databases and papers reported in non-English language was excluded. Experts in the field could have been contacted and asked to provide further published and unpublished studies and the reference lists of all the included studies should have been read. Also, there is no mention of the number of investigators that selected the studies.
Also, I disagree that a comprehensive literature search will always lead to a risk of bias of a review.

Response 3: We added this sentence in the limit of the study

Reviewer 2 Report

I commend the authors for their research entitled "Diagnosis of slipped capital femoral epiphysis: how to stay out of trouble?" The authors focused on the latest evidence concerning the causes of diagnostic delay and risk factors for slipped capital femoral epiphysis (SCFE) and aimed to educate non-orthopaedic physicians (general practitioners and paediatricians) to help reduce delays in diagnosis and provide earlier therapeutic intervention. Overall the topic is interesting, the manuscript is clearly written, and the methods are well presented. Conclusions are based on the results and are nicely illustrated by a Diagnostic algorithm (Figure 2.) However, some points must be cleared before the manuscript could be considered for publication. (1) Results; 3.1.1. – Please define “epiphyseal cupping”. (2) Results; 3.3.1. – Please explain typical radiological features mentioned later in Figure 2 (Trethowan sign – the line of Klein, Southwick’s angle) and show them in a typical X-ray of SCFE here. (3) Discussion; “HOMA” – Please explain every abbreviation when used for the first time through the whole manuscript.  

Author Response

Dear author, thank you for your important suggestions. Next you will find the review point by point.

Point 1:
 (1) Results; 3.1.1. – Please define “epiphyseal cupping”

Response 1: We defined it

Point 2: Results; 3.3.1. – Please explain typical radiological features mentioned later in Figure 2 (Trethowan sign – the line of Klein, Southwick’s angle) and show them in a typical X-ray of SCFE here.

Response 2: We explained all typical radiological features mentioned and we added a typical x-ray of SCFE

Point 3: Discussion; “HOMA” – Please explain every abbreviation when used for the first time through the whole manuscript.  

Response 3: Okay

Round 2

Reviewer 1 Report

The manuscript has been improved.

In the abstract, I do not understand what the authors mean by “
A literature search was conducted in the Science Direct and PubMed databases following the PRISMA guidelines.”. The PRISMA is a guideline for reporting of systematic reviews, not a methodological guideline.

Please explain the exclusion criterion ”articles with low quality” better – what was the cut-point. This can be explained in the section “2.1. Selection criteria” and/or in a method section entitled, e.g., 2.3 Risk of bias assessment.

In the section “2.2 Literature search strategy and data extraction” it has been clarified that two persons independently selected the studies. Please also explain the procedures for the data-extraction.

The PRISMA flow-chard lacks an arrow. The text in the flow chart “Additional records identified through Science Direct” should be replaced with “Records identified through Science Direct”, in my opinion.

Please explain the abbreviations used in the tables after the tables.

In the PRISMA checklist (supplementary material), please note that the page numbers provided may not correspond with the page numbers in the manuscript once it is published. Therefore, I suggest that the authors refer to the sections rather than the page numbers here (e.g., “Section 3.3.”). 
Please don’t leave fields empty in the checklist, but state, “Not applicable”, for example.

In the PRISMA checklist the authors have stated that the Risk of bias in the studies is described on page 3, but I could not find this information anywhere in the manuscript. 

Please review the manuscript for mechanical errors, such as lack of space and misplacement of punctuations.

Author Response

Thank you for your time and important suggestions. I have uploaded the updated version of the manuscript, you will find the changes underlined.
Below the review point by point:

Point 1:
In the abstract, I do not understand what the authors mean by “A literature search was conducted in the Science Direct and PubMed databases following the PRISMA guidelines.”. The PRISMA is a guideline for reporting of systematic reviews, not a methodological guideline.

Response 1: We modified this point.

Point 2: Please explain the exclusion criterion ”articles with low quality” better – what was the cut-point. This can be explained in the section “2.1. Selection criteria” and/or in a method section entitled, e.g., 2.3 Risk of bias assessment

Response 2: We added this part

Point 3: In the section “2.2 Literature search strategy and data extraction” it has been clarified that two persons independently selected the studies. Please also explain the procedures for the data-extraction.

Response 3: We added this part

Point 4: The PRISMA flow-chard lacks an arrow. The text in the flow chart “Additional records identified through Science Direct” should be replaced with “Records identified through Science Direct”, in my opinion.

Please explain the abbreviations used in the tables after the tables.

Response 4: We modified this point

Point 5: In the PRISMA checklist (supplementary material), please note that the page numbers provided may not correspond with the page numbers in the manuscript once it is published. Therefore, I suggest that the authors refer to the sections rather than the page numbers here (e.g., “Section 3.3.”). Please don’t leave fields empty in the checklist, but state, “Not applicable”, for example.In the PRISMA checklist the authors have stated that the Risk of bias in the studies is described on page 3, but I could not find this information anywhere in the manuscript.

Response 5: We modified this point

Point 6: Comments on the Quality of English Language

Please review the manuscript for mechanical errors, such as lack of space and misplacement of punctuations

Response 6: Ok

Kind regards,

Vito Pavone

Reviewer 2 Report

The authors have substantialy improved the manuscript entitled "Diagnosis of slipped capital femoral epiphysis: how to stay out of trouble?" 

Author Response

Thanks for your valuable suggestions, hope to cooperate with you in the future.  Kind regard